# Sliding Mode Observer-Based Stuck Fault and Partial Loss-of-Effectiveness (PLOE) Fault Detection of Hypersonic Flight Vehicle

**Changhua Hu** [1], **Meijie Liu** [1,2], **Hongzeng Li** [1] and **Xiaoxiang Hu** [3,*]

1   204 Unit, Xi'an Research Institute of High-Tech, Xi'an 710025, China
2   Engineering Training Center, Xi'an University of Science and Technology, Xi'an 710054, China
3   School of Automation, Northwestern Polytechnical University, Xi'an 710072, China
*   Correspondence: xxhu@nwpu.edu.cn or huxiaoxiang2008@gmail.com

**Abstract:** In order to improve the safety and reliability of the hypersonic flight vehicle, a sliding mode observer-based fault detection scheme is applied in this paper to handle the actuator fault detection issue, including stuck fault detection and PLOE fault detection. A dynamic linear model with uncertainty is first derived from the original nonlinear hypersonic flight vehicle model by using Taylor's linearization approach at the equilibrium point. Secondly, the actuator fault model, reflecting stuck faults and PLOE faults, is constructed. Then, a sliding mode-based fault detection observer, considering system decomposition, is developed based on the linearized hypersonic flight vehicle model. At last, the designed sliding mode observer is applied to the original nonlinear hypersonic flight vehicle for single-input, single-style actuator fault detection. The simulation results show that stuck faults and big proportion PLOE faults can be timely and accurately detected at the fault time, and the stuck actuator fault from input 3 can cause a deadly impact to the hypersonic flight vehicle, which deserves much more attention than the actuator faults from the other three inputs. Meanwhile, the detection of a small proportion of PLOE faults encounters some difficulties and needs special attention and further investigation.

**Keywords:** hypersonic vehicle; sliding mode observer; actuator fault; fault detection

## 1. Introduction

Much attention has been attracted in both military and civilian hypersonic vehicle (HSV) fields over the last several decades [1,2]. In contrast with ordinary aircrafts, hypersonic vehicles are confronted with more complicated circumstances, such as fast speeds, large payloads, complicated external disturbances and complex environments [3,4]. Based on the fact that tiny, unexpected changes to the hypersonic vehicles may lead to huge flight disasters, the safety and reliability of HSVs have been the focus of numerous researchers. Many effective control methods, including classical and advanced control approaches, have been developed to guarantee the stability of fault-free HSVs [5], such as PID control [6,7], back-stepping control [8,9], sliding mode control [10,11], adaptive control [12,13], predictive control [14,15], etc. However, the above control methods may lose effectiveness, to a large extent, when faults happen in the HSV system. Therefore, fault detection has received significant attention to effectively increase the reliability of the HSV system [16,17].

Model-based fault detection and diagnosis problems have been developed for over 30 years and successfully applied to different fields for system stability and reliability. Model-based extended Kalman filters (EKFs) were used for actuator fault detection of unmanned underwater vehicles [18] and aerial vehicles [19]. Multi-model adaptive estimation methods for actuator fault detection of unmanned aerial vehicles was proposed and simulated in the work of Ducard et al. [20]. Furthermore, an adaptive fault detection observer [21] and a sliding mode observer [22] were designed for actuator fault detection

of near-space vehicles. In the work of Lv et al., the Total Measurable Fault Information Residual (ToMFIR), deriving from an unknown input Kalman filter, was employed for actuator fault detection of a hypersonic flight vehicle [5]. Despite this, there is a lack of relative actuator fault detection works on HSV systems, especially in recent years. To this end, actuator fault detection issues for a new HSV system are the focus of our recent work. Thanks to the sliding mode observer's superior robustness to unknown external uncertainties and nonlinearities [23], it has been widely developed and applied in model-based fault detection fields since its appearance [24,25]. Shen et al. proposed a fault detection method based on a sliding mode observer for a hypersonic vehicle with sensor faults [26]. Alaei et al. applied a new robust H∞ sliding mode observer for state estimation and fault reconstruction of a nonlinear, uncertain boiler system [27]. A sliding mode-based nonlinear fault integrated detector is proposed in the work of Hu et al. for sensor fault detection [28]. To date, there are few studies on actuator fault detection using a sliding mode observer in the HSV system with uncertainty and actuator faults that are carried out.

In addition, as is well known, there are multiple HSV fault styles due to the complexities of the HSV system. Different fault styles mean different generation mechanisms that can cause different impacts on the HSV performance. Therefore, in practice, the HSV faults cannot be simply described as single form, which is generally assumed in previous studies. Moreover, although the sliding mode observer has been successfully applied in fault detection fields, how to employ the sliding mode observer to different fault models in the HSV system is still an issue not yet fully resolved in the sliding mode fields. In this paper, we focus on two different actuator fault models, that is, stuck faults and PLOE faults. A general sliding mode observer for the attitude control system of the HSV with uncertainties and actuator faults is designed for actuator fault detection. A simulation is conducted to validate the effectiveness of the designed sliding mode observer. The results show that the observer in this paper can achieve effective detection and diagnosis for different actuator faults of the HSV.

The rest of this paper is organized as follows: Section 2 states the original nonlinear and linearized HSV model with actuator faults; Section 3 presents the designed sliding mode observer, including its stability and sliding motion reachability proofs; Section 4 implements actuator fault detection simulations on the original nonlinear HSV model to illustrate the effectiveness of the sliding mode observer; and Section 5 gives conclusions and future work directions about the fault detection of the HSV system.

## 2. Problem Formulation

### 2.1. Hypersonic Flight Vehicle Model

In this paper, we introduce a new HSV model. The original nonlinear dynamic equation of the model is expressed, as follows, with system states: $x = \begin{bmatrix} \omega_x & \omega_y & \omega_z & \alpha & \beta & \gamma_v \end{bmatrix}^T$.

$$\begin{cases} \dot{\omega}_x = \frac{M_x + (J_{zz} - J_{yy})\omega_y\omega_z}{J_{xx}} \\ \dot{\omega}_y = \frac{M_y + (J_{xx} - J_{zz})\omega_x\omega_z}{J_{yy}} \\ \dot{\omega}_z = \frac{M_z + (J_{yy} - J_{xx})\omega_x\omega_y}{J_{zz}} \\ \dot{\alpha} = \omega_z + \tan\beta(\omega_y\sin\alpha - \omega_x\cos\alpha) - \frac{Y - G\cos\gamma_v}{mV\cos\beta} \\ \dot{\beta} = \omega_x\sin\alpha + \omega_y\cos\alpha + \frac{Z + G\sin\gamma_v}{mV} \\ \dot{\gamma}_v = \frac{\omega_x\cos\alpha - \omega_y\sin\alpha}{\cos\beta} + \frac{Y\tan\beta - G\tan\beta\cos\gamma_v}{mV} \end{cases} \quad (1)$$

In the above described system equations, the states $\omega_x$, $\omega_y$, $\omega_z$, $\alpha$, $\beta$ and $\gamma_v$ represent the angular rates of rolling, the angular rates of yawing, the angular rates of pitching, the attack angle, the sideslip angle and the velocity inclination, respectively. Meanwhile,

the elements of the control input $u = \delta_c = \begin{bmatrix} \delta_1 & \delta_2 & \delta_3 & \delta_4 \end{bmatrix}^T$ denote the right body flap, left body flap, right rudder and left rudder, which do not appear explicitly in the system equations. The forces $G$, $X$, $Y$ and $Z$ and the moments $M_x$, $M_y$ and $M_z$ are expressed as:

$$
\begin{cases}
G = mg_0 \left( \dfrac{r_0}{r_0 + H} \right)^2 \\[4pt]
X = \frac{1}{2} \rho V^2 S C_x \\[4pt]
Y = \frac{1}{2} \rho V^2 S C_y \\[4pt]
Z = \frac{1}{2} \rho V^2 S C_z \\[4pt]
M_x = \frac{1}{2} \rho V^2 S L m_x \\[4pt]
M_y = \frac{1}{2} \rho V^2 S L m_y \\[4pt]
M_z = \frac{1}{2} \rho V^2 S L m_z
\end{cases}
\tag{2}
$$

With:

$$
\begin{aligned}
C_x =\ & C_x^0 + C_x^{Ma} Ma + C_x^{Ma^2} Ma^2 + C_x^{\alpha} \alpha + C_x^{\alpha^2} \alpha^2 \\
& + C_x^{\alpha \delta_1} \alpha \delta_1 + C_x^{\alpha \delta_2} \alpha \delta_2 + C_x^{\delta_3} \delta_3 + C_x^{\delta_4} \delta_4 \\
C_y =\ & C_y^0 + C_y^{Ma} Ma + C_y^{Ma^2} Ma^2 + C_y^{Ma\alpha} Ma\alpha + C_y^{\alpha} \alpha \\
& + C_y^{\alpha^2} \alpha^2 + C_y^{\delta_1} \delta_1 + C_y^{\delta_2} \delta_2 + C_y^{\delta_3} \delta_3 + C_y^{\delta_4} \delta_4 \\
C_z =\ & C_z^{Ma\beta} Ma\beta + C_z^{\alpha\beta} \alpha\beta + C_z^{\beta} \beta + C_z^{\delta_1} \delta_1 + C_z^{\delta_2} \delta_2 \\
& + C_z^{\alpha \delta_3} \alpha \delta_3 + C_z^{\alpha \delta_4} \alpha \delta_4 \\
m_x =\ & C_{m_x}^{Ma\beta} Ma\beta + C_{m_x}^{\alpha\beta} \alpha\beta + C_{m_x}^{\beta} \beta + C_{m_x}^{\delta_3} \delta_3 + C_{m_x}^{\delta_4} \delta_4 \\
m_y =\ & C_{m_y}^{\alpha\beta} \alpha\beta + C_{m_y}^{\beta} \beta + C_{m_y}^{\delta_1} \delta_1 + C_{m_y}^{\delta_2} \delta_2 + C_{m_y}^{\alpha \delta_3} \alpha \delta_3 \\
& + C_{m_y}^{\alpha \delta_4} \alpha \delta_4 \\
m_z =\ & C_{m_z}^0 + C_{m_z}^{Ma} Ma + C_{m_z}^{Ma\alpha} Ma\alpha + C_{m_z}^{\alpha} \alpha + C_{m_z}^{Ma\delta_1} Ma\delta_1 \\
& + C_{m_z}^{Ma\delta_2} Ma\delta_2 + C_{m_z}^{\alpha \delta_4} \alpha \delta_4 + C_{m_z}^{\alpha \delta_2} \alpha \delta_2
\end{aligned}
\tag{3}
$$

The parameter implications of the model are detailed in Table 1 for a better explanation of the nonlinear HSV model.

**Table 1.** Parameter Implication of the HSV model.

| Symbol | Implication | Symbol | Implication |
|---|---|---|---|
| $m$ | vehicle mass | $X$ | resistance |
| $H$ | altitude | $Y$ | lift |
| $S$ | reference area for dynamic coefficient | $Z$ | lateral force |
| $L$ | reference length for aerodynamic coefficients | $M_x$ | rolling moment |
| $r_0$ | earth radius | $M_y$ | pitching moment |
| $\rho_0$ | mean air density | $M_z$ | yaw moment |
| $g_0$ | acceleration owing to gravity | $\delta_1$ | right body flap |
| $Ma$ | Mach number | $\delta_2$ | left body flap |
| $\gamma_v$ | velocity inclination | $\delta_3$ | right rudder |
| $\alpha$ | attack angle | $\delta_4$ | left rudder |
| $\beta$ | sideslip angle | | |
| $J_{xx}$ | rolling moment of inertia | | |
| $J_{yy}$ | yaw moment of inertia | | |
| $J_{zz}$ | pitch moment of inertia | | |

In this paper, the original nonlinear HSV model is linearized by using Taylor's linearization approach at the equilibrium point for a further fault detection analysis, and the original nonlinear model with uncertainties can be rewritten as:

$$
\begin{aligned}
\dot{x}(t) &= Ax(t) + Bu(t) + D\eta(t, x, u) \\
y(t) &= Cx(t)
\end{aligned}
\tag{4}
$$

where $A \in \Re^{n \times n}$, $B \in \Re^{n \times m}$, $C \in \Re^{p \times n}$ and $D \in \Re^{n \times h}$ are known matrices and the signal $\eta(t, x, u) \in \Re^h$ represents the uncertainties in the system including unmodeled dynamics, external disturbances and parametric uncertainties.

**Remark 1.** *It needs to be pointed out that what we focused on in this paper is the attitude control system of a hypersonic vehicle under the condition of cruise flight. The model of the hypersonic vehicle has high requirements for the response speed of the attitude control system. After a comprehensive analysis, under the condition of cruise flight, the linearized model of the attitude control system has enough of a stability margin. Therefore, the linearized model can satisfactorily be applied for further study under the condition of cruise flight.*

### 2.2. Actuator Fault Model

In this paper, the actuator fault model of the HSV system is formulated as follows:

$$
u = \lambda u_c + f
\tag{5}
$$

where $\lambda = diag\{\lambda_1, \lambda_2, \ldots, \lambda_m\}$ ($m$ is the input dimension) is the unknown constant matrix to be determined, which denotes the extent of the PLOE faults; $f = \begin{bmatrix} f_1 & f_2 & \cdots & f_m \end{bmatrix}^T$ is the stuck fault representing the stuck fault; $u_c$ is the commanded control input for system stability.

Different fault cases can be represented by choosing different values of $\lambda$ and $f$. Two typical fault cases investigated in this paper are described in the next subsections.

#### 2.2.1. Stuck Fault

When $\delta_i$ ($i = 1, 2 \ldots, m$) is stuck, the corresponding fault model can be described as follows:

$$
\begin{cases}
u_i = \lambda_i u_{c,i} + f_i \\
\lambda_i = 0 \\
f_i = f_{s,i}
\end{cases}
\tag{6}
$$

where $f_{s,i}$ denotes the stuck value of $\delta_i$ ($i = 1, 2 \ldots, m$).

#### 2.2.2. PLOE Faults

When the response of the control input cannot be out-and-out, the PLOE faults occur. The PLOE fault model is described as follows:

$$
\begin{cases}
u_i = \lambda_i u_{c,i} + f_i \\
\lambda_i = \lambda_{p,i} \\
f_i = 0
\end{cases}
\tag{7}
$$

where $\lambda_{p,i} \in (0, 1)$ represents the effective ratio of the $i$th control input ($i = 1, 2 \ldots, m$).

Obviously, the system works normally when $\lambda_{p,i} = 1$ and $f_{s,i} = 0$ ($i = 1, 2 \ldots, m$).

### 2.3. Dynamic System Model with Actuator Faults

Based on the aforementioned fault description, we obtained the following HSV system with actuator faults from the linearized dynamic system:

$$
\begin{aligned}
\dot{x}(t) &= Ax(t) + B\lambda u_c(t) + Bf(t) + D\eta(t, x, u_c) \\
y(t) &= Cx(t)
\end{aligned}
\tag{8}
$$

where $A \in \Re^{n \times n}$, $B \in \Re^{n \times m}$, $C \in \Re^{p \times n}$ and $D \in \Re^{n \times h}$ are known matrices ($n > p \geq h$); $x(t) \in \Re^n$ is the state variables; $u_c(t) \in \Re^m$ is the commanded control input by LQR control; $\lambda = diag(\lambda_1, \dots, \lambda_m)$ represents the extent of the PLOE faults; $f(t) \in \Re^m$ is the matrix of the stuck faults; $y(t) \in \Re^p$ is the corresponding measurable output of the system; and the signal $\eta(t, x, u_c) \in \Re^h$ represents the uncertainty in the system.

By defining $g(t) = (\lambda - I)u_c(t) + f(t)$, which reflects the relevant actuator faults, then Equation (8) can be rewritten as follows:

$$\begin{aligned} \dot{x}(t) &= Ax(t) + Bu_c(t) + Bg(t) + D\eta(t, x, u_c) \\ y(t) &= Cx(t) \end{aligned} \tag{9}$$

To realize the SMO design, several assumptions are considered in this paper [24,29].

**Assumption 1.** *The matrices B and C are full rank and $rank(CB) = rank(B)$;*

**Assumption 2.** *$(A, B, C)$ are stale, which means:*

$$rank \begin{bmatrix} SI - A & B \\ C & \mathbf{0} \end{bmatrix} = n + m$$

**Assumption 3.** *$\eta(t, x, u)$ is bounded with $\|\eta(t, x, u)\| \leq \kappa$ for all t, where $\kappa$ is a positive scalar;*

**Assumption 4.** *$g(t) \in \Re^m$ is the bounded fault function to be estimated with $\|g(t)\| \leq \tau(t, u)$.*

**Remark 2.** *It needs to be noted that Assumption 1 and 2 are the necessary and sufficient conditions to guarantee the existence of the sliding motion, which has been proved in previous studies [24,29].*

## 3. Sliding Mode Observer Realizations

### 3.1. Sliding Mode Observer Design

In this subsection, a classical sliding mode observer based on matrix transformation for the linearized HSV model is designed for actuator fault detection as follows:

$$\begin{aligned} \dot{\hat{x}}(t) &= A\hat{x}(t) + Bu_c(t) + L(y - \hat{y}) + Gv(y, \hat{y}) \\ \hat{y}(t) &= C\hat{x}(t) \end{aligned} \tag{10}$$

where $\hat{x}$ and $\hat{y}$ are the estimate of $x$ and $y$, respectively; $G \in \Re^{n \times p}$ is the SMO gain matrix.

The sliding mode discontinuous vector $v$ is designed as follows:

$$v = \begin{cases} \rho(y, u_c, t) \dfrac{P_2 e_y}{\|P_2 e_y\| + \sigma} & if \ e_y \neq \mathbf{0} \\ \mathbf{0} & otherwise \end{cases} \tag{11}$$

where $P_2$ is a symmetric positive-definite (SPD) matrix; $\sigma$ is a small positive scalar; $\rho(y, u_c, t)$ is a scalar function to be determined later; and $e_y$ is defined as $e_y(t) = y(t) - \hat{y}(t)$. Obviously, the realization of the observer design is to determine the parameter $L$, $G$, $\rho(y, u_c, t)$ and $P_2$. The flow chart of the proposed SMO on the attitude control system of the HSV is illustrated in Figure 1.

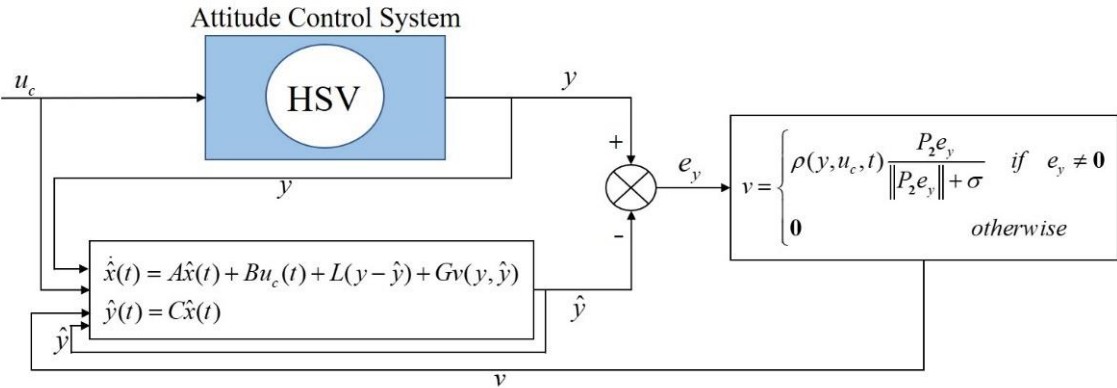

**Figure 1.** Flow chart of the proposed SMO on the attitude control system of the HSV.

Under Assumptions 1 and 2, Equation (9) can be transformed into a concise structure through a change of coordinates $x \rightarrow T_1 x$ as follows:

$$\overline{A} = T_1 A T_1^{-1} = \begin{bmatrix} \overline{A}_{11} & \overline{A}_{12} \\ \overline{A}_{21} & \overline{A}_{22} \end{bmatrix}, \ \overline{B} = T_1 B = \begin{bmatrix} \mathbf{0} \\ \overline{B}_2 \end{bmatrix}, \ \overline{C} = C T_1^{-1} = \begin{bmatrix} \mathbf{0} & I_{p \times p} \end{bmatrix}, \overline{D} = T_1 D = \begin{bmatrix} \overline{D}_1 \\ \overline{D}_2 \end{bmatrix}$$

where $\overline{A}_{11} \in \Re^{(n-p) \times (n-p)}$, $\overline{A}_{12} \in \Re^{(n-p) \times p}$, $\overline{A}_{21} \in \Re^{p \times (n-p)}$, $\overline{A}_{22} \in \Re^{p \times p}$, $\overline{B}_2 \in \Re^{p \times q}$, $\overline{D}_1 \in \Re^{(n-p) \times h}$ and $\overline{D}_2 \in \Re^{p \times h}$. $\overline{B}_2$ has the structure $\overline{B}_2 = \begin{bmatrix} \mathbf{0}_{(p-q) \times q} \\ \overline{B}_0 \end{bmatrix}$, where $rank(\overline{B}_0) = q$.

### 3.2. Stability Analysis of the Designed SMO

Let the estimation error $e(t) = x(t) - \hat{x}(t)$. Thus, the following error system can be obtained from Equations (9) and (10).

$$\begin{aligned} \dot{e}(t) &= \dot{x}(t) - \dot{\hat{x}}(t) = (A - LC)e(t) + D\eta + Bg(t) - Gv \\ e_y(t) &= y(t) - \hat{y}(t) = Ce(t) \end{aligned} \tag{12}$$

**Theorem 1.** *If appropriate matrices $\bar{l}_v$, $P_1$ and $P_2$ are determined to satisfy the following inequations, the sliding motion can be proved asymptotically stable on the sliding surface.*

$$\Phi = \begin{bmatrix} P_1(\overline{A}_{11} + L_0 \overline{A}_{21}) + (\overline{A}_{11} + L_0 \overline{A}_{21})^T P_1 & \overline{A}_{21}^T P_2 & P_1(\overline{D}_1 + L_0 \overline{D}_2) \\ P_2 \overline{A}_{21} & P_2 \bar{l}_v + \bar{l}_v^T P_2 & P_2 \overline{D}_2 \\ (\overline{D}_1 + L_0 \overline{D}_2)^T P_1 & \overline{D}_2^T P & \mathbf{0} \end{bmatrix} < \mathbf{0} \tag{13}$$

$$P_1, P_2 > \mathbf{0}, \left( P_2 \bar{l}_v + \bar{l}_v^T P_2 \right) < \mathbf{0} \tag{14}$$

where $\bar{l}_v$ is a stable SND matrix.

**Proof.** Considering a new coordinate transformation:

$$T_2 = \begin{bmatrix} I_{n-p} & L_0 \\ \mathbf{0} & I_p \end{bmatrix}$$

where $L_0$ is a matrix to be determined.

Therefore, the matrices $(A, B, D, C)$ in Equation (9) can be further transformed to be:

$$\widetilde{A} = T_2 \overline{A} T_2^{-1} = \begin{bmatrix} \widetilde{A}_{11} & \widetilde{A}_{12} \\ \widetilde{A}_{21} & \widetilde{A}_{22} \end{bmatrix}$$

$$= \begin{bmatrix} \overline{A}_{11} + L_0 \overline{A}_{21} & -(\overline{A}_{11} + L_0 \overline{A}_{21})L_0 + \overline{A}_{12} + L_0 \overline{A}_{22} \\ \overline{A}_{21} & -\overline{A}_{21} L_0 + \overline{A}_{22} \end{bmatrix}$$

$$\widetilde{B} = T_2 \overline{B} = \begin{bmatrix} \mathbf{0} \\ \widetilde{B}_2 \end{bmatrix} = \begin{bmatrix} \mathbf{0} \\ \overline{B}_2 \end{bmatrix}, \ \widetilde{C} = \overline{C} T_2^{-1} = \begin{bmatrix} \mathbf{0} & I_{p \times p} \end{bmatrix}$$

$$\widetilde{D} = T_2 \overline{D} = \begin{bmatrix} \widetilde{D}_1 \\ \widetilde{D}_2 \end{bmatrix} = \begin{bmatrix} \overline{D}_1 + L_0 \overline{D}_2 \\ \overline{D}_2 \end{bmatrix}$$

Let $l = \begin{bmatrix} l_1 \\ l_2 \end{bmatrix} = T_2 \overline{L} = T_2 T_1 L$ and $\overline{G} = T_1 G = \begin{bmatrix} -L_0 \\ I_p \end{bmatrix}$, then :

$$\widetilde{G} = T_2 \overline{G} = T_2 \begin{bmatrix} -L_0 \\ I_p \end{bmatrix} = \begin{bmatrix} \mathbf{0} \\ I_p \end{bmatrix}$$

Thus, the error system can be partitioned as:

$$\begin{aligned} \dot{e}_1(t) &= \widetilde{A}_{11} e_1(t) + \left( \widetilde{A}_{12} - l_1 \right) e_2(t) + \widetilde{D}_1 \eta(t, x, u_c) \\ \dot{e}_2(t) &= \widetilde{A}_{21} e_1(t) + \left( \widetilde{A}_{22} - l_2 \right) e_2(t) + \widetilde{D}_2 \eta(t, x, u_c) + \widetilde{B}_2 g(t) - v \end{aligned} \tag{15}$$

The following two Lyapunov function candidates are considered:

$$V_1 = e_1^T P_1 e_1 \tag{16}$$

$$V_2 = e_2^T P_2 e_2 \tag{17}$$

where $P_1 \in \Re^{(n-p) \times (n-p)}$ and $P_2 \in \Re^{p \times p}$ are SPD matrices.

The time derivative of the above Lyapunov functions is obtained as follows:

$$\begin{aligned} \dot{V}_1 &= \dot{e}_1^T(t) P_1 e_1(t) + e_1^T(t) P_1 \dot{e}_1(t) \\ &= \left[ (\overline{A}_{11} + L_0 \overline{A}_{21}) e_1(t) + (\overline{D}_1 + L_0 \overline{D}_2) \eta(t, x, u_c) \right]^T P_1 e_1(t) \\ &\quad + e_1^T(t) P_1 \left[ (\overline{A}_{11} + L_0 \overline{A}_{21}) e_1(t) + (\overline{D}_1 + L_0 \overline{D}_2) \eta(t, x, u_c) \right] \\ &= e_1^T(t) [(\overline{A}_{11} + L_0 \overline{A}_{21})^T P_1 + P_1 (\overline{A}_{11} + L_0 \overline{A}_{21})] e_1(t) \\ &\quad + 2 e_1^T(t) P_1 (\overline{D}_1 + L_0 \overline{D}_2) \eta(t, x, u_c) \end{aligned} \tag{18}$$

$$\begin{aligned} \dot{V}_2 &= \dot{e}_2^T(t) P_2 e_2(t) + e_2^T(t) P_2 \dot{e}_2(t) \\ &= \left[ \overline{A}_{21} e_1(t) + l_v e_2(t) + \overline{D}_2 \eta(t, x, u_c) + \overline{B}_2 g(t) - v \right]^T P_2 e_2(t) \\ &\quad + e_2^T(t) P_2 \left[ \overline{A}_{21} e_1(t) + l_v e_y(t) + \overline{D}_2 \eta(t, x, u_c) + \overline{B}_2 g(t) - \overline{B}_2 v \right] \\ &= e_2^T(t) (P_2 l_v + l_v^T P_2) e_2(t) + 2 e_2^T(t) P_2 \overline{A}_{21} e_1(t) \\ &\quad + 2 e_2^T(t) P_2 \overline{D}_2 \eta(t, x, u_c) + 2 e_2^T(t) P_2 \overline{B}_2 g(t) - 2 e_2^T(t) P_2 v \end{aligned} \tag{19}$$

Then, we can obtain the summation of the two derivatives:

$$\begin{aligned} \dot{V} &= \dot{V}_1 + \dot{V}_2 \\ &= e_1^T(t) [(\overline{A}_{11} + L_0 \overline{A}_{21})^T P_1 + P_1 (\overline{A}_{11} + L_0 \overline{A}_{21})] e_1(t) \\ &\quad + 2 e_1^T(t) P (\overline{D}_1 + L_0 \overline{D}_2) \eta(t, x, u_c) - 2 e_2^T(t) P_2 v \\ &\quad + e_2^T(t) (P_2 l_v + l_v^T P_2) e_2(t) + 2 e_2^T(t) P_2 \overline{A}_{21} e_1(t) \\ &\quad + 2 e_2^T(t) P_2 \overline{D}_2 \eta(t, x, u_c) + 2 e_2^T(t) P_2 \overline{B}_2 g(t) \end{aligned} \tag{20}$$

Substituting Equation (11) into the above equation, the following inequality is obtained:

$$2 e_2^T(t) P_2 \overline{B}_2 g(t) - 2 e_2^T(t) P_2 v \leq 2 \| P_2 e_2 \| (\| \overline{B}_2 \| \tau - \rho(y, u_c, t)) \tag{21}$$

Thus, $\dot{V}$ can be rewritten as:

$$
\begin{aligned}
\dot{V} \leq\; & e_1{}^T(t)[(\overline{A}_{11} + L_0\overline{A}_{21})^T P_1 + P_1(\overline{A}_{11} + L_0\overline{A}_{21})]e_1(t) - 2\rho_0\|P_2e_2\| \\
& + 2e_1{}^T(t)P(\overline{D}_1 + L_0\overline{D}_2)\eta(t, x, u_c) + 2e_2{}^T(t)P_2\overline{D}_2\eta(t, x, u_c) \\
& + e_2{}^T(t)(P_2l_v + l_v{}^T P_2)e_2(t) + 2e_2{}^T(t)P_2\overline{A}_{21}e_1(t)
\end{aligned}
\tag{22}
$$

The inequation can be transformed to the following structure:

$$
\dot{V} \leq \begin{bmatrix} e_1 \\ e_2 \\ \eta \end{bmatrix}^T \Phi \begin{bmatrix} e_1 \\ e_2 \\ \eta \end{bmatrix} - 2\rho_0\|P_2e_2\|
\tag{23}
$$

where:

$$
\Phi = \begin{bmatrix}
P_1(\overline{A}_{11} + L_0\overline{A}_{21}) + (\overline{A}_{11} + L_0\overline{A}_{21})^T P_1 & \overline{A}_{21}{}^T P_2 & P_1(\overline{D}_1 + L_0\overline{D}_2) \\
P_2\overline{A}_{21} & P_2\bar{l}_v + \bar{l}_v{}^T P_2 & P_2\overline{D}_2 \\
(\overline{D}_1 + L_0\overline{D}_2)^T P_1 & \overline{D}_2{}^T P & 0
\end{bmatrix}
$$

Obviously, if the matrix $\Phi$ satisfies $\Phi < 0$, $\dot{V} < 0 - 2\rho_0\|P_2e_2\| < -2\rho_0\|P_2e_2\|$.

Finally, the asymptotically stability of the designed SMO is proofed, and $\Phi < 0$ is a sufficient condition for it. Meanwhile, the SMO can exhibit excellent robustness to the uncertainty signal $\eta$.

Proved. □

### 3.3. Reachability of the Sliding Motion

An appropriate value of $\rho(y, u_c, t)$ in Equation (11) can successfully drive the error system to the sliding surface $S$ in finite time, which can ensure the stability of the designed SMO.

**Theorem 2.** *If the parameter $\rho$ in Equation (11) is chosen as follows, the sliding motion on $S = [e : \overline{C}e = 0]$ can be reached and remain on it.*

$$
\rho \geq \|\overline{A}_{21}\|\Theta + \|\overline{B}_2\|\tau + \|\overline{D}_2\|\kappa + \rho_0
\tag{24}
$$

**Proof.** The Lyapunov function Equation (17) is considered here, and its derivative along the trajectory is Equation (18). We can easily obtain the following inequality:

$$
\dot{V}_2 \leq e_2{}^T(t)(P_2\bar{l}_v + \bar{l}_v{}^T P_2)e_2(t) + 2\|P_2e_2\|(\|\overline{A}_{21}\|\|e_1\| + \|\overline{D}_2\|\kappa + \|\overline{B}_2\|\tau - \rho)
\tag{25}
$$

It is known that $P_2\bar{l}_v + \bar{l}_v{}^T P_2 < 0$; thus, the selection of the value of $\rho$ plays a decisive role on the sliding motion reachability. There exists an instant $t_l$ and a positive scalar $\Theta$ such that $\|e_1(t)\| \leq \Theta, \forall t \geq t_l$. Based on the above information, a proper $\rho$ can be selected by the inequation of $\rho \geq \|\overline{A}_{21}\|\Theta + \|\overline{B}_2\|\tau + \|\overline{D}_2\|\kappa + \rho_0$; thus, the following inequality is obtained:

$$
\dot{V}_2 \leq -2\rho_0\|P_2e_2\|, \forall t \geq t_l
\tag{26}
$$

Proved. □

## 4. Simulation Result

In this section, the SMO design method in Section 3 is applied to the linearized HSV model to derive the SMO observer. The linearized system model with uncertainty is shown as follows:

$$\dot{x}(t) = Ax(t) + Bu_c(t) + Bg(t) + D\eta(t, x, u_c)$$
$$y(t) = Cx(t) \tag{27}$$

where:

$$A = \begin{bmatrix} 5.000 & 1.000 & 0 & 0 & -57.2098 & 1.000 \\ 1.000 & 2.000 & 0 & 3.7897 & 0 & 0.500 \\ 0 & 0 & 1.000 & -0.0116 & -0.594 & 0 \\ 0 & 1.000 & 0 & -0.0183 & 0 & 0 \\ 0.1676 & 0 & 0.9859 & 0 & -0.0029 & 0.0020 \\ 0.9859 & 0 & -0.1676 & 0 & -0.0005 & 0 \end{bmatrix}$$

$$B = \begin{bmatrix} 0 & 0 & -24.1144 & -42.0982 \\ -26.8390 & 7.8990 & 0 & 0 \\ 0.0235 & 0.0135 & -0.3168 & -4.4629 \\ -0.0116 & -0.0016 & 0 & 0 \\ 0 & 0 & -0.0501 & -0.3009 \\ 0.0200 & 0.0300 & 0 & 0 \end{bmatrix}, \ C = \begin{bmatrix} 1 & 0 & 0 & 0 & 0 & 0 \\ 0 & 1 & 0 & 0 & 0 & 0 \\ 0 & 0 & 1 & 0 & 0 & 0 \\ 0 & 0 & 0 & 1 & 0 & 0 \end{bmatrix}$$

$$D = \begin{bmatrix} 0 \\ 0 \\ 0 \\ 0.1 \\ 0 \\ 0 \end{bmatrix}, \ \eta(t, x, u_c) = 0.01\sin(t) \text{ and } u_c(t) = -kx(t).$$

The parameter $k$ is determined by LQR control as follows:

$$k = \begin{bmatrix} -0.8272 & 2.5867 & 7.9792 & -55.2867 & 0.4948 & 184.6327 \\ -1.5978 & 5.9769 & 16.8881 & 139.5611 & -11.9185 & 733.3548 \\ -7.7928 & -0.0224 & 55.2191 & 0.4697 & 51.0559 & -7.6711 \\ -272.2053 & 0.1350 & -137.5423 & 0.3875 & -370.3044 & -124.3171 \end{bmatrix}$$

Obviously, Assumptions 1–3 are satisfied. The transformation matrix $T_1$ is computed as follows:

$$T_1 = \begin{bmatrix} 0.0352 & 0.4342 & 1.4120 & -1.3215 \times 10^3 & -25.8657 & -185.4388 \\ 0.0053 & 0.4342 & 0.2120 & -1.3231 \times 10^3 & -3.8835 & -184.9869 \\ 1 & 0 & 0 & 0 & 0 & 0 \\ 0 & 1 & 0 & 0 & 0 & 0 \\ 0 & 0 & 1 & 0 & 0 & 0 \\ 0 & 0 & 0 & 1 & 0 & 0 \end{bmatrix}$$

The transformed matrices are described as follows:

$$\overline{A} = \begin{bmatrix} 0.1208 & -0.1221 & 186.5527 & -1.3206 \times 10^3 & 6.8459 & 23.7877 \\ 0.0147 & -0.0159 & 182.5693 & -1.3223 \times 10^3 & 27.3696 & 24.2492 \\ 2.6046 & -2.6164 & 4.9222 & 1.0051 & -3.1231 & -19.8739 \\ 4.7772 \times 10^{-4} & -0.0032 & 1 & 2.0012 & 0 & 0.2111 \\ 0.0259 & -0.0260 & -7.7463 \times 10^{-4} & 2.7487 \times 10^{-5} & 0.9689 & -0.1382 \\ 0 & 0 & 0 & 1 & 0 & -0.0183 \end{bmatrix}$$

$$\overline{B} = \begin{bmatrix} 0 & 0 & 0 & 0 \\ 0 & 0 & 0 & 0 \\ 0 & 0 & -24.1144 & -42.0982 \\ -26.8390 & 7.8990 & 0 & 0 \\ 0.0235 & 0.0135 & -0.3168 & -4.4629 \\ -0.0116 & -0.0016 & 0 & 0 \end{bmatrix}, \overline{C} = \begin{bmatrix} 0 & 0 & 1 & 0 & 0 & 0 \\ 0 & 0 & 0 & 1 & 0 & 0 \\ 0 & 0 & 0 & 0 & 1 & 0 \\ 0 & 0 & 0 & 0 & 0 & 1 \end{bmatrix}, \overline{D} = \begin{bmatrix} 0.4342 \\ 0.4342 \\ 0 \\ 1 \\ 0 \\ 0 \end{bmatrix}$$

The sliding mode observer is designed as introduced in Section 3. The parameters $\rho$ and $\sigma$ are chosen as $\rho = 0.01$ and $\sigma = 0.001$, respectively. The relative gains from the observer representation, determined by solving Equation (13) and (14), are as follows:

$$\overline{l}_v = \begin{bmatrix} -3.2523 & -0.0019 & -0.0274 & 0 \\ -0.0019 & -0.5000 & 0 & 0 \\ -0.0274 & 0 & -0.5003 & 0 \\ -0.0040 & -0.0094 & -0.0008 & -9.46 \times 10^3 \end{bmatrix}$$

$$L_0 = \begin{bmatrix} -0.3451 & 369.3912 & 1.4706 \times 10^{-7} & 1.3215 \times 10^3 \\ 0.2242 & 5.9882 \times 10^{-4} & 1.8286 \times 10^{-6} & 1.3231 \times 10^3 \end{bmatrix}$$

$$T_2 = \begin{bmatrix} 1 & 0 & -0.3451 & 369.3912 & 1.4706 \times 10^{-7} & 1.3215 \times 10^3 \\ 0 & 1 & 0.2242 & 5.9882 \times 10^{-4} & 1.8286 \times 10^{-6} & 1.3231 \times 10^3 \\ 0 & 0 & 1 & 0 & 0 & 0 \\ 0 & 0 & 0 & 1 & 0 & 0 \\ 0 & 0 & 0 & 0 & 1 & 0 \\ 0 & 0 & 0 & 0 & 0 & 1 \end{bmatrix}$$

$$G = \begin{bmatrix} 1 & 0 & 0 & 0 \\ 0 & 1 & 0 & 0 \\ 0 & 0 & 1 & 0 \\ 0 & 0 & 0 & 1 \\ -0.0505 & 33.6227 & 0.0546 & -0.2223 \\ 0.0035 & -0.7035 & 0 & 7.1572 \end{bmatrix}$$

$$L = \begin{bmatrix} 8.1745 & 1.0070 & -3.0957 & -19.8739 \\ 1.0019 & 2.5012 & 1.8883 \times 10^{-5} & 0.2111 \\ 0.0266 & 4.6359 \times 10^{-5} & 1.4692 & -0.1382 \\ 0.0040 & 1.0094 & 7.9173 \times 10^{-4} & 9.4600 \times 10^3 \\ 16.8053 & 41.7998 & 1.0937 & 4.0535 \\ 0.6467 & 6.2774 & -0.1731 & -0.2405 \end{bmatrix}$$

In the end, the sliding mode observer is successfully obtained, which is further applied to the original nonlinear HSV system. The simulation of the observer without actuator faults is carried out as illustrated in Figure 2, and the system is stabilized by LQR control. The states $w_z$ and $\alpha$ had steady-state errors caused by the uncertain part $\eta(t, x, u_c)$. It is obvious that the system states are successfully observed through the sliding mode observer. In order to study the observer performance for separate actuator faults, single-input, single-style actuator faults are considered in this section, whereas multi-input, multi-style actuator fault combinations are left for further study.

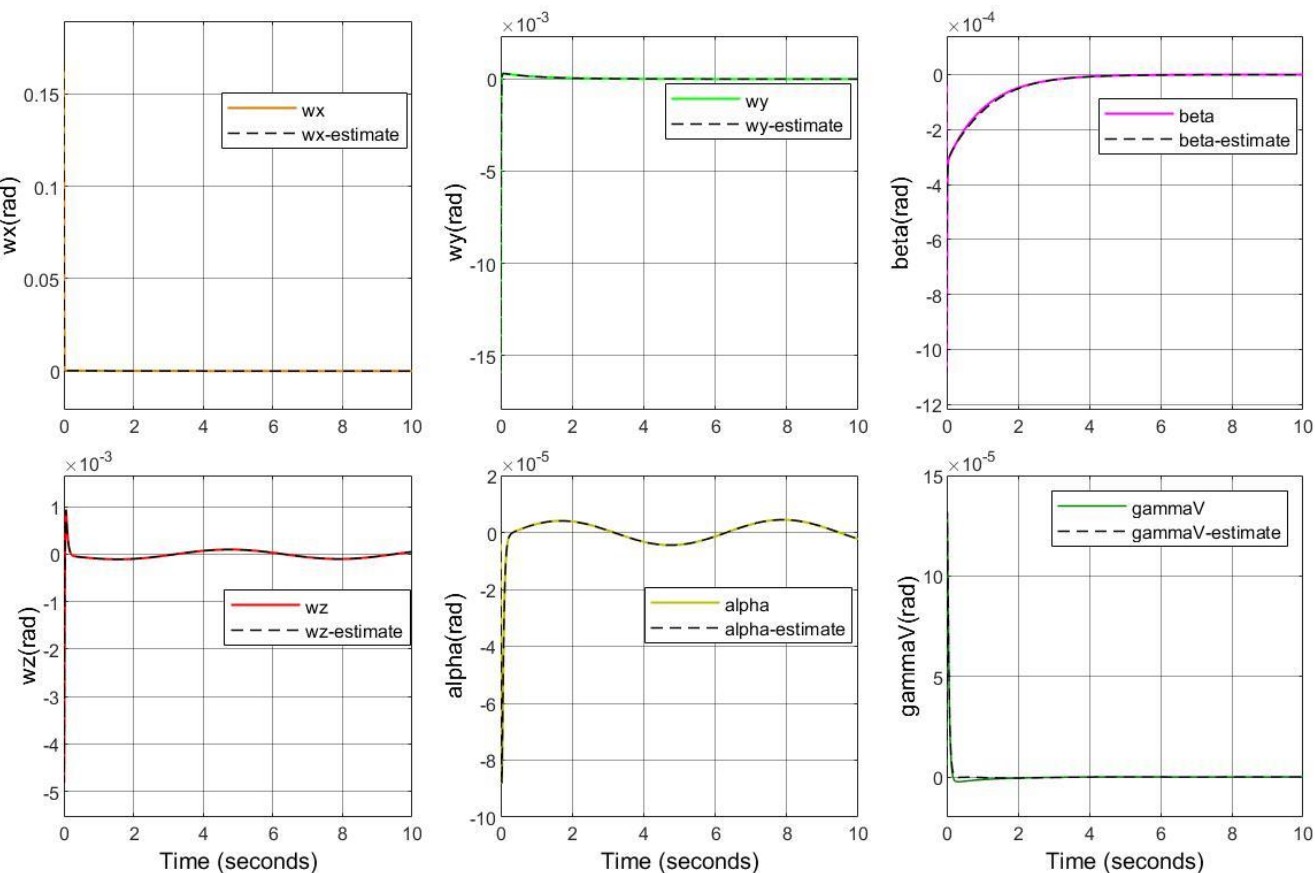

**Figure 2.** State estimation by the sliding mode observer without actuator faults.

### 4.1. Simulation Results of Stuck Fault Detection

Considering the fact that the original nonlinear HSV system has been stabilized by the LQR feedback control and all four inputs approach zero (see Figure 3), any small stuck fault means huge challenges for the controlled HSV system. Therefore, the small stuck fault value from four inputs is chosen as $u_i = 0.006$ ($i = 1, 2, 3, 4$). Assume that the actuator stuck faults occur at $5$ s $< t < 10$ s . The actuator fault detection simulation results from four inputs are illustrated in Figures 4–7, respectively. It is noted that system states exhibit a sudden change at $t = 5$ s , and the actuator faults can be detected immediately. Meanwhile, it is obvious that the designed SMO performs a robust state observation even when stuck faults occur.

As illustrated in Figures 4 and 5, the stuck faults from input 1 and input 2 cannot be obviously detected through the observation of the states $w_x$ and $w_y$, but can be reflected from the observation of the other four states independently. Therefore, the detection of stuck faults from input 1 and 2 can be successfully achieved through the observation of the states $w_z$, $\alpha$, $\beta$ and $\gamma_v$.

For the stuck faults from inputs 3 and 4, a sudden change happened in each state at $t = 5$ s , as shown in Figures 6 and 7. It is worth noting that the extent of the sudden change brought by the stuck fault from input 3 is much more severe than those from the other three, and exhibits no convergence tendency, which is detailed in Figure 8. Thus, stuck actuator faults from input 3 may cause a deadly impact on the control of the HSV system and should be paid more attention to avoid possible accidents. In general, the stuck faults that occurred in the HSV model can be accurately and timely detected through the designed sliding mode observer, and some stuck faults may result in severe HSV disasters.

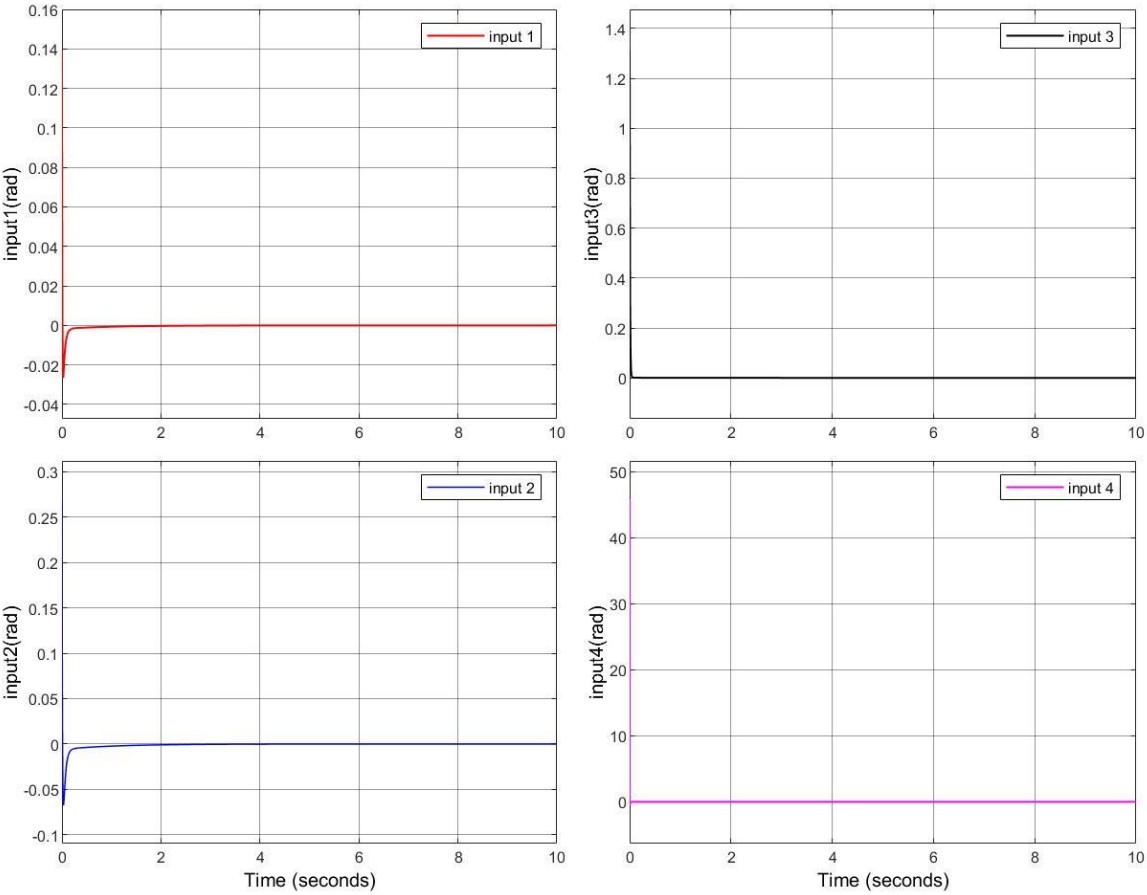

**Figure 3.** Simulation of inputs 1–4 without actuator faults.

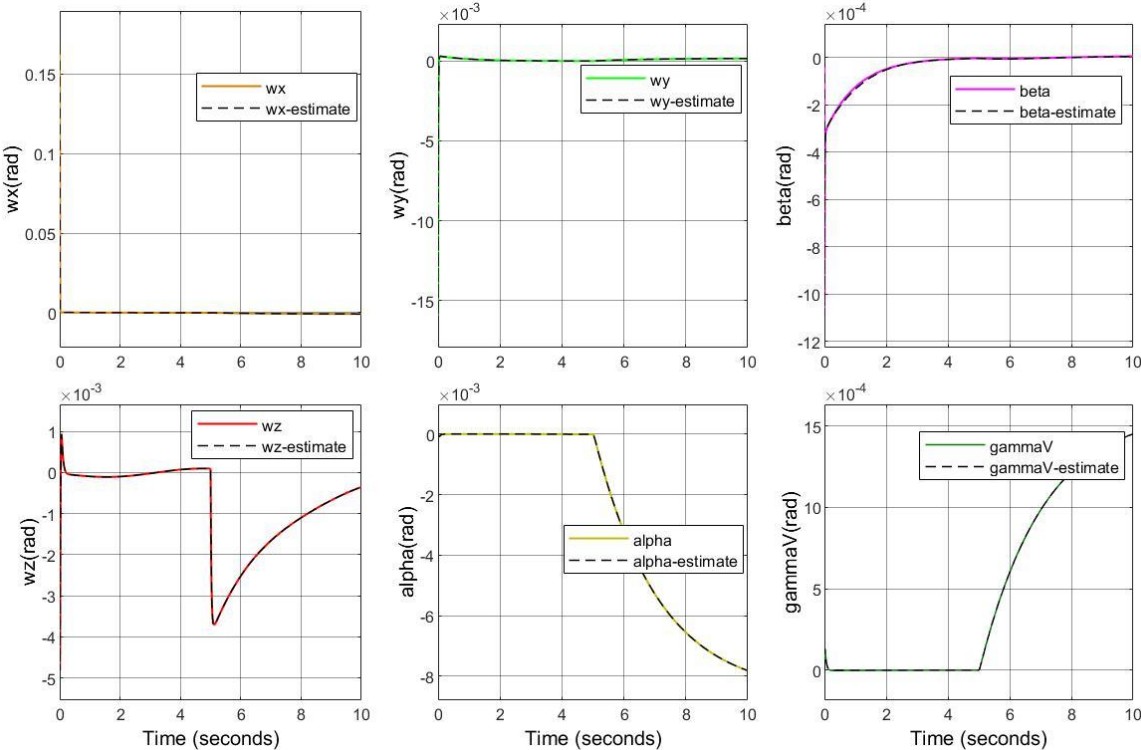

**Figure 4.** Detection simulation results of actuator stuck fault from input 1.

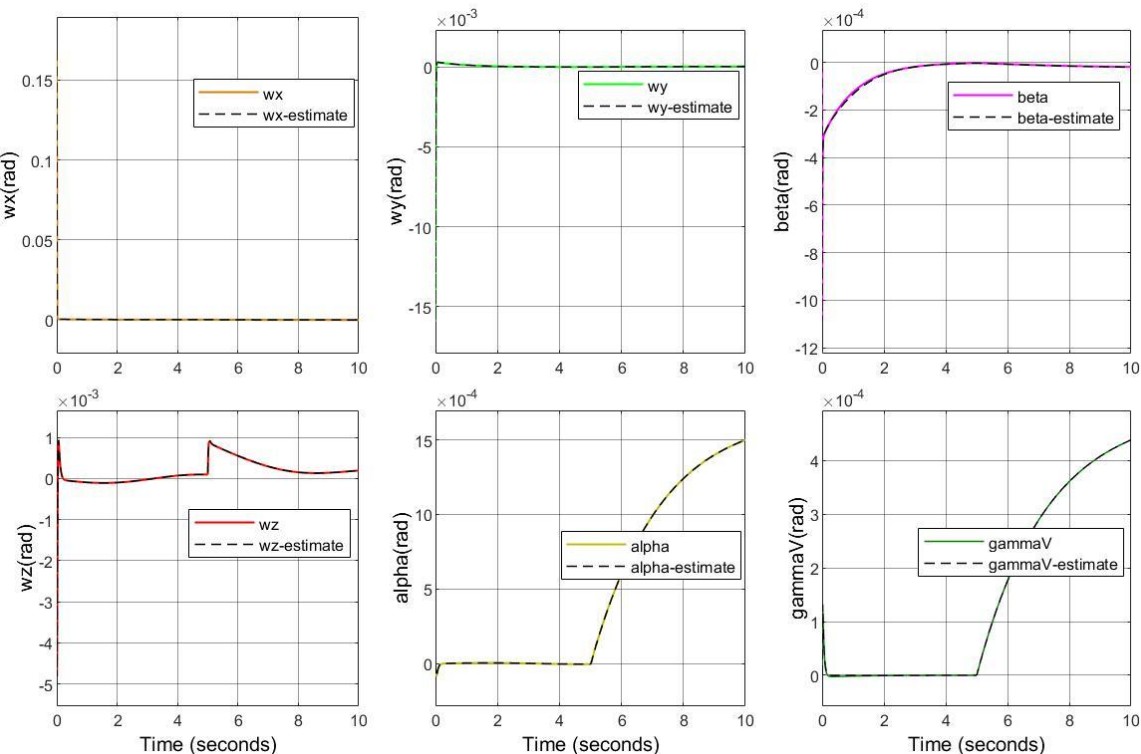

**Figure 5.** Detection simulation results of actuator stuck fault from input 2.

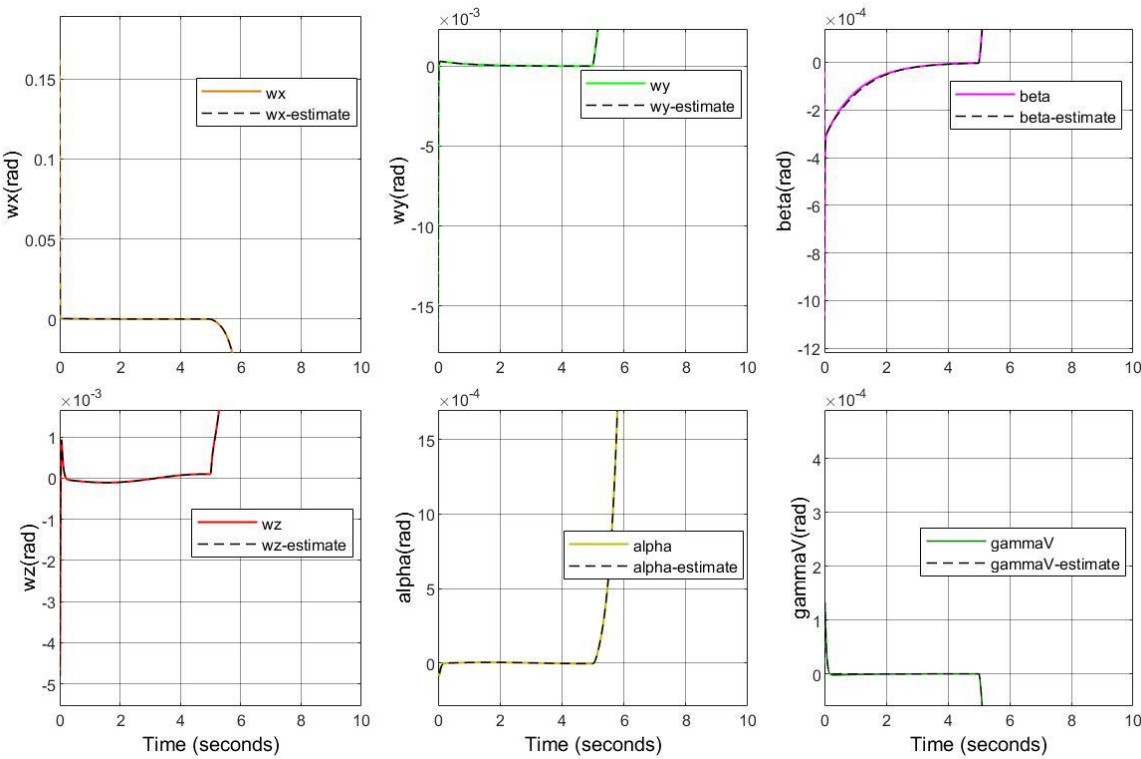

**Figure 6.** Detection simulation results of actuator stuck fault from input 3.

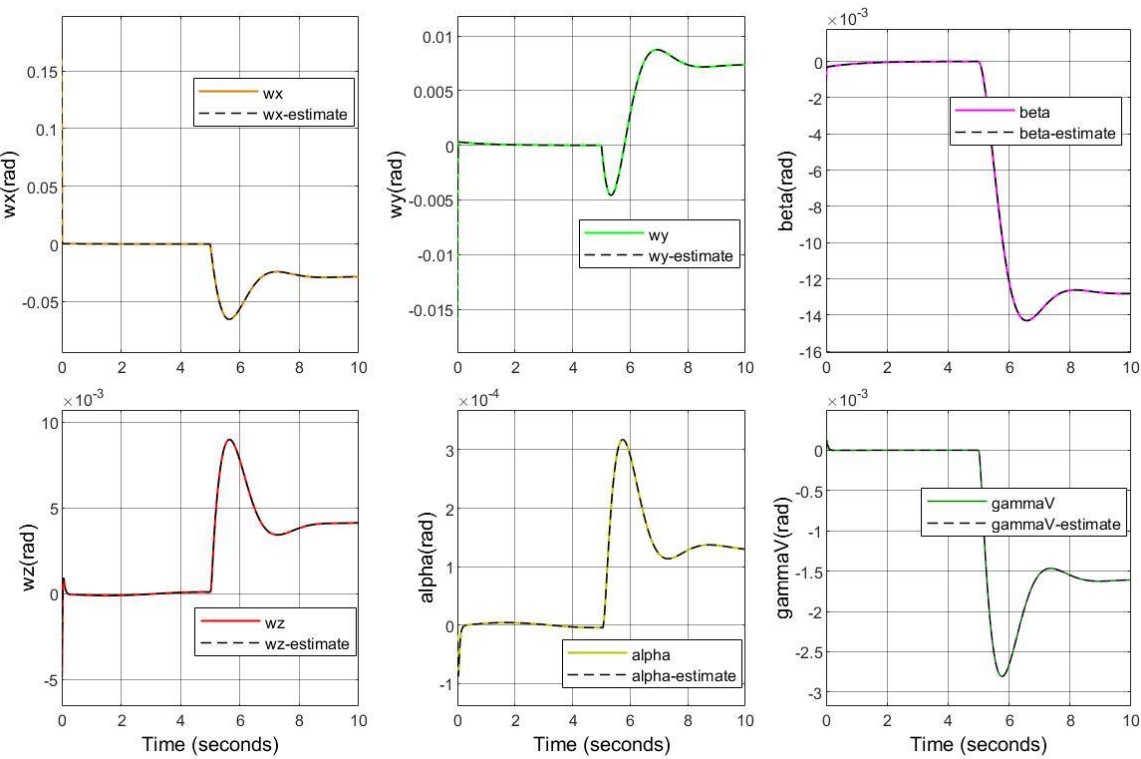

**Figure 7.** Detection simulation results of actuator stuck fault from input 4.

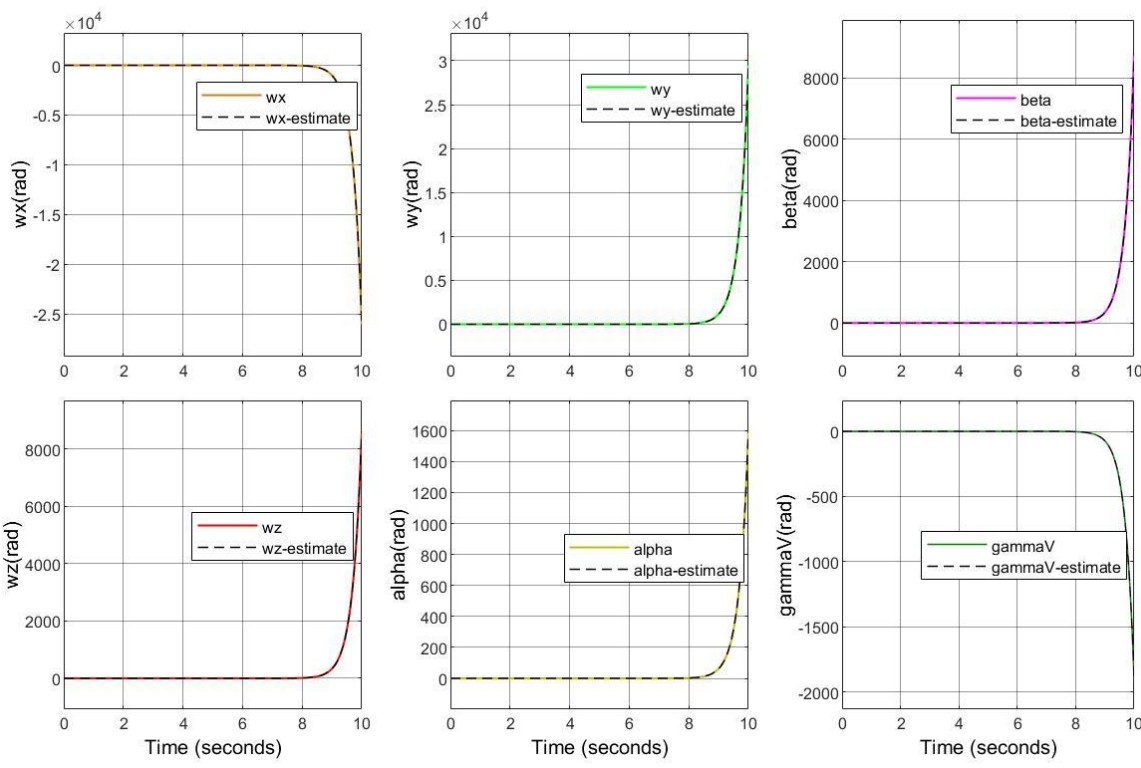

**Figure 8.** Detection simulation results of actuator stuck fault from input 3.

### 4.2. Simulation Results of PLOE Fault Detection

The stuck faults are reported to be well detected through the designed sliding mode observer in the above subsection, whereas the PLOE fault detection encounters some difficulties in this subsection. Four extents of PLOE faults are detected through the sliding mode

observer, that is, 80%, 60%, 40% and 20% PLOE faults. As described in Section 2, the PLOE faults are represented as $u_i = \lambda_i u_{c,i}$ ($i = \mathbf{1}, \mathbf{2}, \mathbf{3}, \mathbf{4}$) and $\lambda_i =$ 20%, 40%, 60%, 80% denoted 80%, 60%, 40% and 20% PLOE faults, respectively. Because of similar simulation results of the PLOE fault from the four independent inputs, the PLOE fault detection simulation from input 2 is exhibited in this subsection as a representative. The detection simulation of the abovementioned four PLOE faults from input 2 is illustrated in Figures 9–12, with the faults occurring at $\mathbf{5}\,\text{s} < t < \mathbf{10}\,\text{s}$.

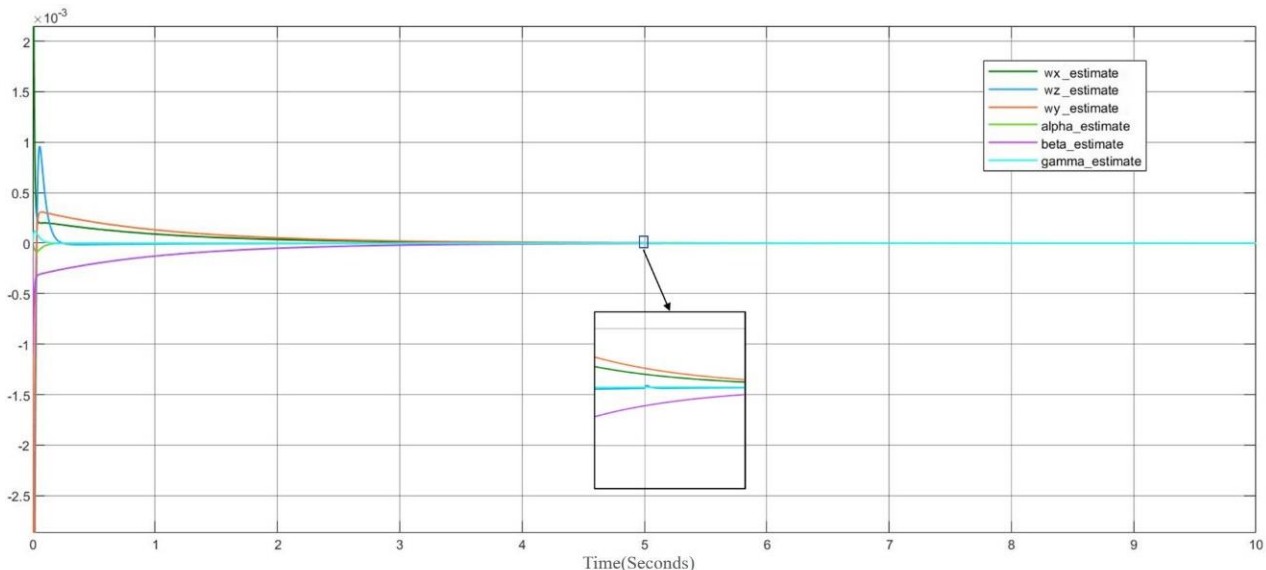

**Figure 9.** Input 2's 20% PLOE fault detection simulation.

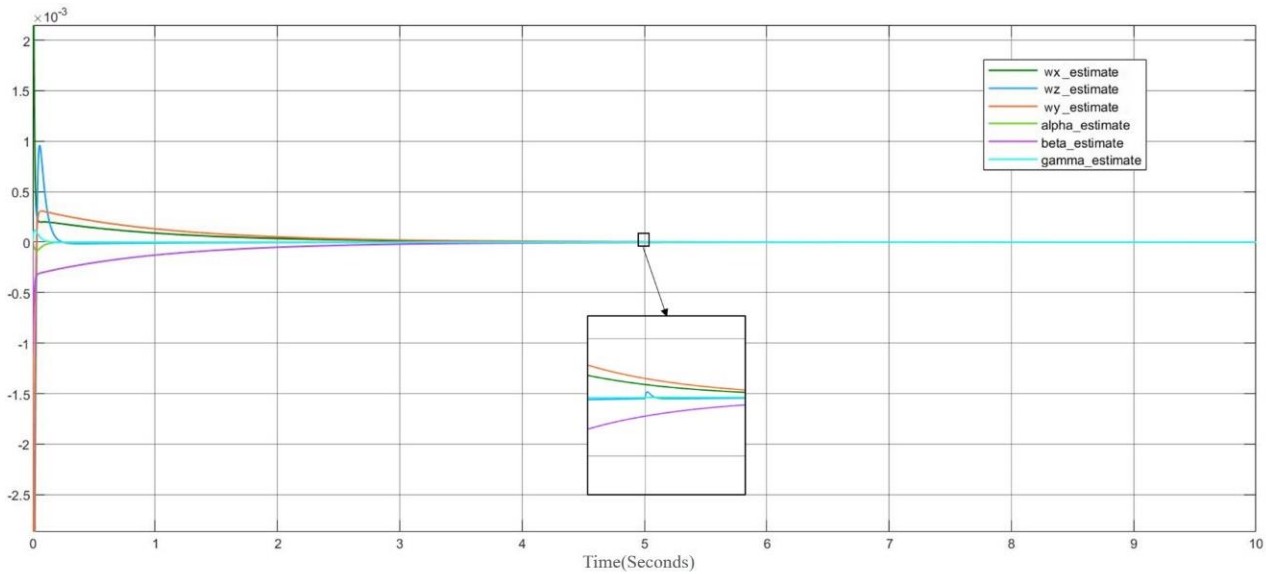

**Figure 10.** Input 2's 40% PLOE fault detection simulation.

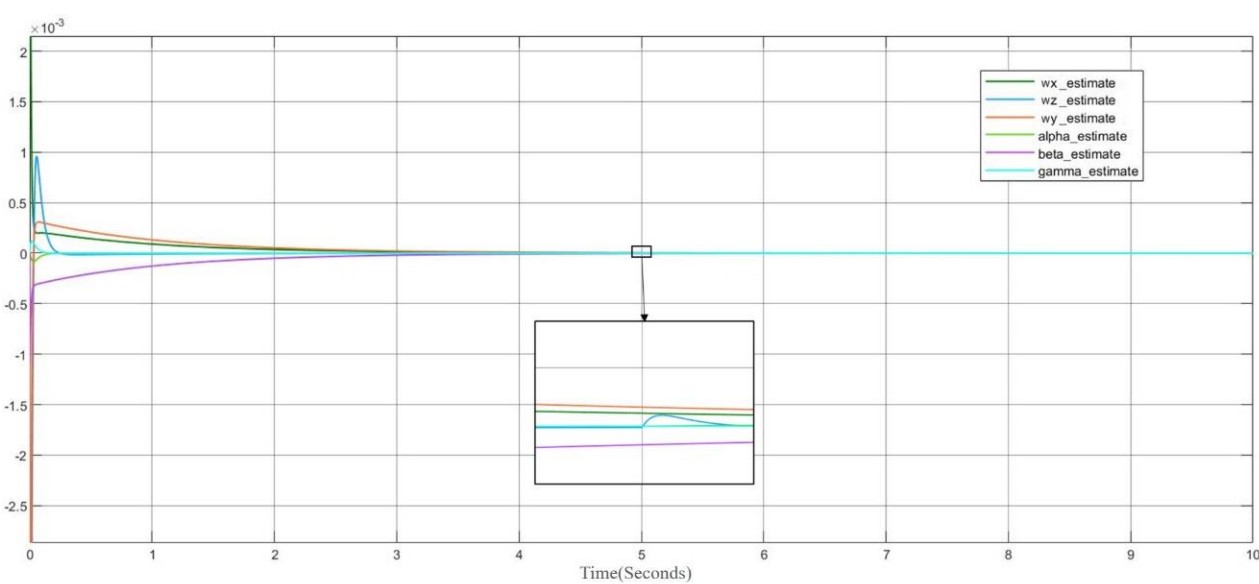

**Figure 11.** Input 2's 60% PLOE fault detection simulation.

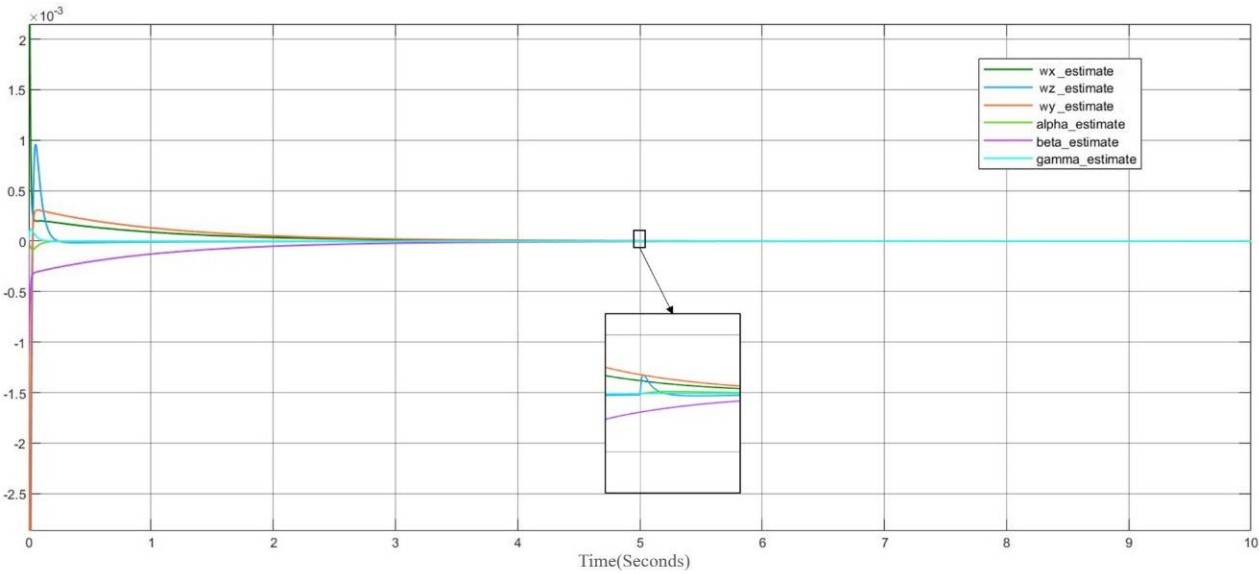

**Figure 12.** Input 2's 80% PLOE fault detection simulation.

Based on the fact that the system is stabilized by LQR feedback control when no faults occur, the four inputs all converge to zero in finite time, as shown in Figure 3. Thus, the value of inputs has become extremely small at the fault time. Moreover, closed-loop systems have been known as robust systems for external disturbances, including faults [30,31]. Thus, even though the PLOE faults occur, a small proportion of PLOE faults lead to tiny changes to the system, which are hardly detected owing to both the tiny change value and the existence of feedback control. From Figures 9–12, it is obviously found that the HSV system exhibits basic stabilities when the PLOE faults occur. For example, under the 20% PLOE fault from input 2 in Figure 9, only the state $w_z$ exhibits a sudden tiny change that lasts an extremely short time, which is hardly detected, whereas the other five states have no change. The amplitude of the state change at the fault time increases with the increase in the PLOE proportion, which is obviously shown in Figures 9–12. In other words, the bigger the proportion, the better the fault detection effectiveness. However, the disadvantage is that the detection may miss a low proportion of PLOE faults.

## 5. Conclusions and Future Work Direction

### 5.1. Conclusions

A sliding mode observer is considered in this paper to carry out single-input, single-style actuator fault detection for a HSV model. The HSV model is first linearized at an equilibrium point to develop the model-based fault detection observer. Then, the designed sliding mode observer is applied to the original HSV system. Stuck faults with a value of 0.006 at $5$ s $< t < 10$ s from four inputs are added to the system separately. From the simulation results, it is found that stuck faults can be immediately and successfully detected through the sliding mode observer in spite of the existence of uncertainty. In particular, all system states rapidly diverge from the fault moment under the stuck fault from input 3. Therefore, more attention should be paid to input 3 in order to avoid deadly impact on the HSV system. For the PLOE faults, the sliding mode observer can successfully detect a big proportion of PLOE faults, but it encounters some difficulties in a small proportion of PLOE fault detections, which needs further investigation.

### 5.2. Future Work Direction

A sliding mode observer is applied to a HSV model for single-input, single-style actuator fault detection in this paper. Most of the actuator faults can be well detected. There still exists much work to follow:

1.  For the possibility of misdetection of a small proportion of PLOE faults, the sliding mode observer or another suitable observer needs to be further improved to solve the misdetection problem;
2.  The linear system for the observer design can be improved by a TS-fuzzy method to approach the original nonlinear system for more precise detection;
3.  After the actuator fault detection, further efforts, such as fault construction, should be made to achieve fault tolerant control and guarantee the system's stability.

**Author Contributions:** Conceptualization, C.H. and M.L.; methodology, X.H.; software, M.L.; validation, C.H., H.L. and M.L.; investigation, X.H.; resources, C.H.; data curation, M.L. and H.L.; writing—original draft preparation, M.L.; writing—review and editing, C.H.; visualization, X.H.; supervision, X.H.; project administration, H.L.; funding acquisition, C.H. and X.H. All authors have read and agreed to the published version of the manuscript.

**Funding:** This research was funded by the National Natural Science Foundation of China under Grant 61833016, Grant 62073265.

**Acknowledgments:** We would like to acknowledge the reviewers for their careful reading, helpful comments, and constructive suggestions, which have significantly improved the presentation of our manuscript.

**Conflicts of Interest:** The authors declare no conflict of interest.

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
