# Peer review of "Sliding Mode Observer-Based Stuck Fault and Partial Loss-of-Effectiveness (PLOE) Fault Detection of Hypersonic Flight Vehicle"

_electronics, doi:10.3390/electronics11193059_

Round 1

Reviewer 1 Report

1. The feasibility of the proposed sliding mode solution resides in the capacity of the synthesis of real-world physical implementations. What circuitry would be associated with such a solution? At least a flowchart detailing should be presented.

2. The vehicle's flight characteristics and dynamics are not presented, therefore is quite difficult to assess that a linear approximation approach would be satisfactory.

3. In more detail: Is Taylor's linearization enough to control the fast dynamics of the HSV? On which grounds/criteria one can assess that the linearized model is satisfactory?

4. What is the meaning of the expression "relatively happened" (Section II, page 3, right column)?

5. How did the authors choose the numerical values within matrices A, B, C, D, T1, etc.? How are these values connected to the vehicle's characteristics, trajectory, attitude, supposed flight atmospheric conditions, modeled fault type . etc?

6. What are the considered characteristics of the uncertainty signal η?

Author Response

Dear reviewer:

The co-authors and I would like to thank you for the time and effort spent in reviewing the manuscript. We also would like to thank you for your careful reading, helpful comments, and constructive suggestions, which has significantly improved the presentation of our manuscript. In the past several days, we have revised our manuscript seriously according to your valuable comments and suggestions, and each comment will be directly addressed regarding the modified manuscript with changes colored in red.

Reviewer 2 Report

The paper is devoted to sliding mode observer design for a hypersonic vehicle system with two different types of faults. The paper is clear, well-written and may be considered for publication in Electronics journal after short revision (see, the list of comments below). Also, it's worth discussing if the obtained observation has been compared with the existing ones (not only sliding mode) and what benefits it gives.

1. Page 1: "Many effective control methods, including classical and advanced control approaches, have been developed to guarantee the stability of fault-free HSV[5], such as PID control, back-stepping control, sliding mode control, adaptive control, and predictive control, etc." Please, add citations on the listed control methods.

2. Page 1: explain the meaning "ToMFIR".

3. In equation (4), explain, please, how the output (matrix C) is chosen.

4. Under equation (8), the dimension q is wrong, it should be changed to h.

5. Add physical sense of assumptions I and II, please.

6. $e_y$ in equation (12) is not defined.

7. In (25), is $\tau$ the same as in assumption IV?

8. In section IV, the values of gain coefficient k are not given.

9. At the end of the theorem should be written " proved".

Author Response

(The authors gave the same response as above.)

Round 2

Reviewer 1 Report

The authors have responded satisfactorily to my remarks and concerns. Consequently, the manuscript has been significantly improved. I would just make a final recommendation. The manuscript represents engineering research, thus specifying the units of measurement is an important aspect. In that respect, I would also recommend specifying the unit of measurement as axes labels for all the plotted dependence graphs. If the plotted quantities are normalized or dimensionless it should be explicitly specified. 

Author Response

The authors have responded satisfactorily to my remarks and concerns. Consequently, the manuscript has been significantly improved. I would just make a final recommendation. The manuscript represents engineering research, thus specifying the units of measurement is an important aspect. In that respect, I would also recommend specifying the unit of measurement as axes labels for all the plotted dependence graphs. If the plotted quantities are normalized or dimensionless it should be explicitly specified. 
